# Cell Culture Media, Unlike the Presence of Insulin, Affect α-Synuclein Aggregation in Dopaminergic Neurons

**DOI:** 10.3390/biom12040563

**Published:** 2022-04-09

**Authors:** Irena Hlushchuk, Justyna Barut, Mikko Airavaara, Kelvin Luk, Andrii Domanskyi, Piotr Chmielarz

**Affiliations:** 1Institute of Biotechnology, HiLIFE, University of Helsinki, Viikinkaari 5D, 00790 Helsinki, Finland; irena.hlushchuk@helsinki.fi; 2Drug Research Program, Faculty of Pharmacy, University of Helsinki, Viikinkaari 5E, 00014 Helsinki, Finland; mikko.airavaara@helsinki.fi; 3Department of Brain Biochemistry, Maj Institute of Pharmacology, Polish Academy of Sciences, Smętna 12, 31-343 Kraków, Poland; jbarut@if-pan.krakow.pl; 4Neuroscience Center, HiLIFE, University of Helsinki, Haartmaninkatu 8, 00014 Helsinki, Finland; 5Department of Pathology and Laboratory Medicine, Center for Neurodegenerative Disease Research, Perelman School of Medicine, University of Pennsylvania, Philadelphia, PA 19104, USA; kelvincl@pennmedicine.upenn.edu

**Keywords:** dopaminergic neuron, alpha-synuclein, lewy body, pathological protein aggregation, insulin, Parkinson’s disease

## Abstract

There are several links between insulin resistance and neurodegenerative disorders such as Parkinson’s disease. However, the direct influence of insulin signaling on abnormal α-synuclein accumulation—a hallmark of Parkinson’s disease—remains poorly explored. To our best knowledge, this work is the first attempt to investigate the direct effects of insulin signaling on pathological α-synuclein accumulation induced by the addition of α-synuclein preformed fibrils in primary dopaminergic neurons. We found that modifying insulin signaling through (1) insulin receptor inhibitor GSK1904529A, (2) SHIP2 inhibitor AS1949490 or (3) PTEN inhibitor VO-OHpic failed to significantly affect α-synuclein aggregation in dopaminergic neurons, in contrast to the aggregation-reducing effects observed after the addition of glial cell line-derived neurotrophic factor. Subsequently, we tested different media formulations, with and without insulin. Again, removal of insulin from cell culturing media showed no effect on α-synuclein accumulation. We observed, however, a reduced α-synuclein aggregation in neurons cultured in neurobasal medium with a B27 supplement, regardless of the presence of insulin, in contrast to DMEM/F12 medium with an N2 supplement. The effects of culture conditions were present only in dopaminergic but not in primary cortical or hippocampal cells, indicating the unique sensitivity of the former. Altogether, our data contravene the direct involvement of insulin signaling in the modulation of α-synuclein aggregation in dopamine neurons. Moreover, we show that the choice of culturing media can significantly affect preformed fibril-induced α-synuclein phosphorylation in a primary dopaminergic cell culture.

## 1. Introduction

Parkinson’s disease (PD) is the second most prevailing neurodegenerative disorder, with an estimated 7 to 10 million PD patients worldwide. Its economic burden, both on carriers of the disease and health care systems, is heavy [1,2]. PD is one of several neurodegenerative disorders that feature Lewy bodies (LB) as a pathophysiological hallmark.

The major component of LB and Lewy neurites (LN) had been considered to be the misfolded alpha-synuclein (α-syn) [3,4]. Recently, a more complex nature of LB/LN was described [5], elegantly demonstrating the presence of both aggregated proteins and membranous material, originating from vesicles and organelles [5]. Remarkably, only a small amount of such inclusions featured a primarily filamentous structure, though the presence of filamentous α-syn in LB is supported by other studies [6]. The majority of α-syn deposited in LB is phosphorylated at the serine 129 (Ser129) residue and is easily detectable by specific antibodies.

The potential of insulin in the central nervous system (CNS) has been studied with genuine enthusiasm because of its pleiotropic effects on neurons. Insulin has been reported to modulate memory, learning, neuronal survival [7,8,9], neuronal plasticity [10,11], cognitive functions and even longevity [12]. In patients, insulin resistance is correlated with age-related neurodegenerative diseases, such as dementia-associated diabetes mellitus, Alzheimer’s disease [13] and PD [11,14,15]. Prospective therapies with insulin and its sensitizer, for example, in Alzheimer’s disease, have been under investigation in recent years [16,17]. Naturally, the importance and incredible range of actions of insulin in the CNS, along with the prospect of its positive effect on neurodegenerative disorders, are motivation to further pursue the investigation of the mechanisms of insulin’s action in the CNS.

Insulin is primarily synthesized in pancreatic beta cells of Langerhans islets and enters the CNS through the blood–brain barrier by carrier-mediated, saturable, carefully regulated transport. The mediators regulating this transport are yet to be discovered and described, but the existing load of evidence suggests that inflammation, diabetes and triglycerides enhance insulin transport to the CNS, while aging, obesity, fasting and Alzheimer’s disease weaken it [18,19]. Numerous studies suggest the production of insulin not only in the pancreas but also locally in the brain (reviewed in [18,20]). Regardless of the origin, insulin has different effects in different tissues. In the peripheral tissues, such as the liver, muscle and adipose tissue, insulin’s main role is the regulation of glucose uptake; in the brain, however, the insulin effect is considered neurotrophic [20]. Insulin orchestrates many cellular processes, including gene expression regulation [21,22], but its main functions are currently considered to be the stimulation of glucose, protein and lipid metabolism.

Insulin’s actions are mediated by insulin receptors (IRs), a disulfide-linked (αβ)2 transmembrane dimer of heterodimers that form a subfamily of receptor tyrosine kinases, along with the insulin-like growth factor 1 (IGF-1) receptor and the insulin receptor-related receptor [23]. The cells expressing IRs and IGFRs can also form hybrid receptors with unknown physiological roles. These hybrid receptors are composed of an IR αβ-heterodimer and an IGF1R αβ-heterodimer and have a lower affinity to insulin as compared to IGF-1/IGF-2 [24,25,26,27].

Insulin’s presumable potential to enhance neuronal survival is based on its negative modulation of the pro-apoptotic protein expression via the PI3K-AKT signaling pathway. J. G. Mielke and Y. T. Wang [28] reported the neuroprotective effect of insulin in cultured rat hippocampal neurons, deprived of oxygen and glucose. M. Ramalingam and S-J. Kim [29] described the neuroprotective role of insulin in 1-Methyl-4-phenyl pyridinium (MPP+)-induced toxicity in retinoic acid-differentiated human neuroblastoma SH-SY5Y cells. They demonstrated the dose-dependent manner of insulin’s effect and suggested that it may be due to the PI3K-AKT survival pathway activation that inhibits MPP+-induced iNOS and ERK activation. We have recently demonstrated that Glial Cell Line-Derived Neurotrophic Factor (GDNF), which acts through RET kinase, is capable of activating the PI3K-AKT pathway and reducing the accumulation of aggregated α-syn with the possible involvement of PI3K-AKT [30]. AKT can enhance the degradation of uptaken misfolded α-syn in endocytic organelles [31]. The similarities in intracellular signaling pathways activated by insulin and GDNF receptor RET prompted us to consider whether a similar protective effect may be exerted by insulin treatment.

To study the pathology and test various treatment approaches, numerous models that represent different aspects of PD are exploited. Existing models are divided into animal models (neurotoxin-based and genetic) [32,33,34] and cellular models (differentiated progenitor cells, primary midbrain culture and midbrain-like organoids) [35,36,37,38]. Current models of PD only partially reproduce the disease, mainly because the causes provoking neurodegeneration in sporadic PD are yet to be discovered. For research to be efficient, such partial models need to develop the pathology and be easy to quickly and reliably manipulate and monitor.

The choice of a model for a study greatly depends on the aspect of the disease and/or the kind of therapy the study is focused on. For investigating the signaling pathways and testing new compounds, a cellular model is a safe first choice. In comparison to animal models, cellular models are typically less expensive and can be used to screen a larger number of compounds in a shorter time [37].

In this study, we modeled the LB pathology (α-syn-containing inclusions) in the cell culture of primary dopaminergic neurons by adding exogenous α-syn preformed fibrils (PFFs). Exogenous α-syn PFFs recruit soluble endogenous α-syn of the neuron to aggregate and promote the development of the insoluble LN-like and LB-like inclusions within neurites and neuronal perikarya, respectively [39,40]. These aggregates progressively develop over a span of several days, first in the neurites and, after about a week, also in the perikarya. The cell death in this model does not occur during the time of the experiment, but only after prolonged incubation [41].

An important and often overlooked factor in the design of studies on primary neurons is a composition of commonly utilized media supplements containing various growth factors. In fact, we have recently demonstrated that neurotrophic factors included in the media supplements may have a profound impact on modeling the LB pathology in primary dopaminergic cultures [30]. In the current study, we set out to investigate whether the presence or absence of insulin in media formulations affects the development of the LB pathology. To this end, we have tested different media formulations, as well as compounds modulating insulin signaling pathways. We found no evidence of direct effects of insulin on α-syn aggregation in primary dopaminergic neurons. However, we found large differences in the extent of the accumulation of phosphorylated α-syn aggregates observed in neurons cultured in different media formulations, which may have an impact on the design and interpretation of other studies.

## 2. Materials and Methods

### 2.1. Primary Mouse Embryonic Neuronal Cultures

The LB-like pathology in the primary neuronal culture was modeled by the administration of exogenously prepared α-syn PFFs [42]. The primary midbrain cell culture preparation was based on the assay developed by A. Planken and colleagues [43] and optimized by S. Er and colleagues [38].

Briefly, the midbrain area was dissected from embryonic day 13.5 (E13.5) NMRI mouse embryos, washed with HBSS, dissociated by incubation with trypsin for 20 min at 37 °C and subsequently treated with DNase I in HBSS containing 50% FBS. The viability and number of dissociated cells were quantified, and the cells were plated in micro-islands at a density of 35,000 cells/well on 96-well plates. Hippocampal and cortical cultures were prepared in a similar way from embryonic day 16–17 (E16–17) CD1 mouse embryos and plated at 50,000 cells per well in 384-well plates.

The cells were cultured either in:Dopamine neuron medium (DPM) (Dulbecco’s Modified Eagle’s Medium DMEM/F12 (Thermo Scientific (Gibco) #21331–020,Waltham, MA, USA), 5 µM L-Glutamine (Thermo Scientific (Gibco), #25030–032, Waltham, MA, USA), 1× N-2 serum supplement (Thermo Scientific, #17502–048, Waltham, MA, USA), 150 µM D-glucose (Sigma-Aldrich, #G8769, St. Louis, MO, USA) and 200 ng/mL Primocin (Invivogen; ant-pm-1, ant-pm-2, San Diego, CA, USA));Neurobasal Medium cocktail (NB) (Neurobasal Medium (-) L-Glutamine (Thermo Scientific (Gibco), #21103-049, Waltham, MA, USA), 1× B-27 Supplement (Thermo Scientific (Gibco), #17504-044, Waltham, MA, USA), 1.25 µM L-Glutamine (Thermo Scientific (Gibco), #25030–032, Waltham, MA, USA) and 200 ng/mL Primocin (Invivogen; ant-pm-1, ant-pm-2, San Diego, CA, USA));Neurobasal Medium cocktail *sans* insulin (NB-ins), essentially similar to NB, but having 1× B-27 Supplement Minus Insulin (Thermo Scientific (Gibco), #A18956-02, Waltham, MA, USA).

The media were filtered with a syringe filter FP 30/0.2, pore size 0.2 μM (Sigma-Aldrich, #10462200, St. Louis, MO, USA) and used on the day of preparation.

### 2.2. Compounds and Treatments

To induce the formation of α-syn aggregates, emblematic for LB pathologies, on the day in vitro (DIV), 8 dopaminergic neurons were treated with α-syn PFFs, diluted in 1× PBS to concentration 100 μg/mL and sonicated at high power with a Bioruptor sonication device (Diagenode, Liege, Belgium, #B01020001), 10 cycles, 30 s on/30 s off, to achieve fibril fragment lengths ~50 nM [38]. For experiments on cortical or hippocampal neurons, commercially available mouse α-syn PFFs (StressMarq, SPR-324, Cadboro Bay, Victoria, BC, Canada) were diluted in 1× PBS to a final concentration of 100 µg/mL and sonicated with a high-power probe sonicator (UP100H, Hielscher, Teltow, Germany) with the following settings: 2 mM sonicator probe, 60 s with maximum power pulses, 0.5 s on/0.5 s off on ice. The final concentration of PFFs per well for all experiments was 2.5 μg/mL.

As a positive control, we chose glial cell line-derived neurotrophic factor (GDNF), due to its ability to consistently reduce α-syn accumulation of the LB pathology in primary mouse embryonic midbrain cultures exposed to α-syn PFFs [30,44]. Recombinant GDNF (PeproTech, #450-44, London, UK) was added at a final concentration of 50 ng/mL. GDNF was added either simultaneously with compound treatment on DIV12 or 15 min before adding PFFs on DIV8 in the experiment with different media on midbrain cultures.

To study the effect of different compounds on developing α-syn aggregates, compounds were added to the cell culture on DIV12. The chosen compounds were diluted in Dimethyl Sulfoxide (DMSO) according to their solubility information to create stock solutions with concentrations of 20 or 10 mM. The prepared solutions were aliquoted and frozen at −80 °C. Immediately before the experiment, the compound stock solutions were thawed, diluted in fresh DPM to obtain the desired concentrations and added to the cells in quantities of 1.5 μL per well. To minimize possible position effects, the positioning of the wells on the plate for each treatment group was evenly balanced.

The following compounds targeting insulin signaling were utilized, as shown in the Figure 1.

GSK1904529A (Selleckchem #S1093, Munich, Germany)—IGF1R and IR selective inhibitor, reversibly and ATP-competitively inhibits ligand-induced phosphorylation of IGF-1R and IR at concentrations above 0.01 μM, followed by blocking downstream signaling. M. WT. 851.96. Soluble in DMSO 170 mg/mL (199.54 mM); IC50 = 27 nM. Tested concentrations in the well were 250 nM, 25 nM and 2.5 nM.

AS 1949490 (Tocris, #3718, Bristol, UK)—SHIP2 inhibitor, increases AKT phosphorylation specifically related to insulin but not with growth factor treatment (Suwa et al. 2009). M. WT. 371.88. Solubility in DMSO 100 mM, max concentration 37.19 mg/mL; IC50 = 0.34 μM. Tested concentrations in the well were 5 μM, 1 μM, 0.5 μM and 0.05 μM.

VO-OHpic (Tocris, #3591, Bristol, UK)—PTEN inhibitor, potently and selectively inhibits PTEN over cysteine-based phosphatases and increases PIP2 and PIP3 levels. M. WT. 361.16. Solubility in DMSO 100 mM, max concentration 36.12 mg/mL; IC50 = 35 nM. Tested concentrations in the well were 250 nM and 25 nM.

On DIV15, the plates were fixed with 4% paraformaldehyde (PFA; Sigma-Aldrich, #158127, St. Louis, MO, USA) in PBS for 20 min and immunostained with anti-phoshoSer129-α-synuclein (rabbit anti-pSer129-α-syn, Abcam, #ab51253, 1:2000) and either sheep polyclonal anti-TH (Merck (Millipore), #AB1542, 1:2000, Darmstadt, Germany) or mouse monoclonal anti-NeuN (Abcam, ab104224, 1:500, Cambridge, UK).

The mechanism and effect of α-syn phosphorylation at Ser129 are still under debate [45]. Nevertheless, pathologically aggregated α-syn has been shown to be consistently phosphorylated at Ser129, thus providing a good and convenient marker for detecting α-syn aggregates. In a previous study, we have validated such an approach through staining with other markers of aggregated α-syn [30].

GSK1904529A and AS 1949490 were tested in the same plates and utilized the same negative (VEH) and positive (GDNF) control groups. However, they are represented on separate graphs for clarity of presentation.

### 2.3. Imaging and Analysis

All the plates were scanned with the automated microscope for well-plates, Image Xpress Nano Automated Imaging System (Molecular Devices, LLC, San Jose, CA, USA), with three fluorescent filters (dopaminergic neurons) or TCS SP8 WLL (Leica, Wetzlar, Germany) in widefield mode (cortical and hippocampal neurons). Acquired images of each well were analyzed with the CellProfiler, and the CellProfiler Analyst software packages [46] adapted as previously described by us [38].

### 2.4. Statistical Analysis

In vitro data from the independent experiments (plates) were analyzed with a randomized block ANOVA design, matching groups from different plates [47]. Statistical significance was calculated by mixed-model one-way or two-way analysis of variance (ANOVA), followed by a Holm–Sidak multiple comparison test. GraphPad Prism version 9.3.0 for Windows (GraphPad Software, San Diego, CA, USA, www.graphpad.com) was used to perform all statistical tests.

The statistical significance threshold was set at *p* < 0.05. The experimental data presented are mean ± SD.

## 3. Results

### 3.1. Pharmacological Modulation of Insulin Signaling Does Not Affect α-Syn Aggregation

The DPM culturing media for primary dopaminergic cultures contains an N2 supplement containing insulin. Therefore, to investigate the role of insulin in PFF-induced α-syn aggregation, we blocked the insulin receptor by adding GSK1904529A in concentrations of 250 nM, 25 nM and 2.5 nM on DIV12 to the primary midbrain culture grown in DPM and treated with PFFs on DIV8. As a positive control on DIV12, we used GDNF.

GSK1904529A is a selective inhibitor of IGF-1R and IR that blocks the receptors’ autophosphorylation and downstream signaling. The results of four independent experiments showed no significant effect of the compound (Figure 2), while the positive control group showed a clear reduction in the α-syn aggregation (*p* < 0.0001).

To investigate whether enhancing insulin signaling through the PI3K-AKT would affect the accumulation of pαSyn in dopaminergic neurons, we selectively inhibited negative regulators of this pathway, PTEN or SHIP2, by treating primary embryonic midbrain culture with VO-OHpic (Figure 3) or AS1949490 (Figure 4), respectively.

Treatment with 25 and 250 nM PTEN inhibitor VO-OHpic showed no effect, either on the accumulation of pαSyn in TH-positive neurons (Figure 3A) or on the survival of PFFs-treated dopaminergic neurons (Figure 3B).

Soeda and colleagues [45], in their study, observed an impairment of insulin’s neuroprotective effect in corticogeniculate neurons when SHIP2 was overexpressed. They also identified a memory impairment in SHIP2 transgenic mice and demonstrated the improvement of impaired synaptic plasticity and memory in diabetic mice with SHIP2 inhibitor AS1949490 [48].

Our results showed no significant effect of AS1949490 on the formation of LB-like aggregates in dopaminergic neurons for concentrations from 0.05 µM to 1 µM (Figure 4A). AS1949490 in a concentration of 5 μM shows a significant decrease in TH-positive cell numbers (Figure 4B), making interpretation of its effects on α-syn accumulation at this concentration unfeasible (Figure 4A,B).

### 3.2. Culture Media Affect α-Syn Aggregation in Dopaminergic Neurons

Since neither the inhibition of IR/IGF-1R nor the facilitation of signaling through the PI3K/AKT pathway downstream to these receptors affected PFF-induced α-syn aggregation, we decided to investigate whether a complete withdrawal of insulin could affect this process.

DPM contains 5 mg/L insulin (recombinant full chain) originating from the N2 supplement, which contains 500 mg/L of insulin. The most appropriate medium for comparison would be insulin-free DMEM/F12 and N2 supplement; however, such reagents are not commercially available. Due to these limitations, we decided to change culturing media to an alternative commonly used formulation, which also ensures the survival of dopaminergic neurons—NB medium. The main component of the NB cocktail—Neurobasal Medium (-) L-Glutamine—lacks insulin in the formulation. However, another component of the NB cocktail, the B27 Supplement, can be found both with and without insulin, for example, Gibco B27 Supplement Serum-Free, in whose description insulin concentration is declared as confidential, and its insulin-free counterpart Gibco B27 Supplement Minus Insulin (without insulin or antioxidant cocktail in the description). To support neuronal survival immediately after preparation of primary cultures, we plated the cells in DPM, as recommended for the midbrain culture [38,43], with the routine partial media change on DIV2, as described below. On DIV5, however, we gently washed each well with 150 μL of warm Phosphate-Buffered Saline (PBS) and then added 150 μL of freshly prepared DPM, NB or NB-ins media according to the treatment groups.

The analysis of the images from three independent repeats with two-way repeated-measures ANOVA revealed significant effects of the medium (*p* < 0.05), GDNF (*p* < 0.05) and their interaction (*p* < 0.01). The following post hoc multiple comparison tests revealed that the cell culture, maintained in NB or NB-ins from DIV5 to DIV15, showed significantly fewer pαSyn aggregates in TH-positive cells in comparison with the cell culture maintained in DPM (*p* < 0.001, Figure 5B). The effect of the GDNF on the accumulation of α-syn was significantly greater in TH-positive neurons maintained in DPM, compared to those in NB and NB-ins (*p* < 0.01 for medium–GDNF interaction in two-way repeated-measure ANOVA), albeit GDNF remained effective in both NB and NB-ins media. The insulin present in the NB media, however, failed to significantly influence the accumulation of α-syn in dopaminergic neurons, compared to the insulin-deprived NB-ins (Figure 5A,B).

TH-positive cells of the midbrain culture maintained in NB and NB-ins tended to survive better than those maintained in DPM (Figure 5C).

To investigate further, we tested whether a variation of the culture media would have a similar effect in other neurons, such as primary hippocampal and cortical neurons. As expected, the incidence of α-syn aggregates in hippocampal neurons was almost three times higher compared to cortical neurons. However, media variation had no significant effect (Figure 6A,B).

## 4. Discussion

The main objective of the study was to investigate the role of the insulin signaling pathways on the intracellular accumulation of phosphorylated α-syn.

### 4.1. Formation of Intracellular α-Synuclein Accumulation in Dopaminergic Neurons Treated with a Selective Inhibitor of IGF-1R and IR

The neuroprotective effect of insulin was described in several in vitro models, e.g., PD-model in retinoic acid-differentiated human neuroblastoma SH-SY5Y cells with MPP(+)-induced neurotoxicity [29] and H_2_O_2_ toxicity [49], and models of ischemia (oxygen–glucose deprivation) in the rat hippocampal neurons and primary cortical neurons [49,50]. IGF-1 has also been studied as a neuroprotective agent in PD [51].

Since insulin is present in the N2 supplement added to the DPM medium, to evaluate the effect of the insulin on the formation of intracellular α-syn aggregates, we treated dopaminergic neurons with selective IGF-1R/IR inhibitor GSK1904529A. This small-molecule kinase inhibitor belongs to imidazopyridines and potently and selectively blocks IGF-1R and IR receptors’ autophosphorylation and downstream signaling. Sabbatini and colleagues [52] determined the half-maximal inhibitory concentration of GSK1904529A for IR and IGF-1R at 25 nM and 27 nM, respectively, establishing the potency of this compound. They have also described the antitumor effect of GSK1904529A in tumor xenograft models in vitro. This effect has been interpreted as the result of blocking the PI3K/AKT and MAPK pathways in cells that lead to the transcriptional downregulation of the regulators of the G1-phase in the cell cycle.

Our results, however, showed no significant change in the survival of the dopaminergic neurons upon blocking the IR/IGF-1R with GSK1904529A. Similar to the results on the survival rate, GSK1904529A showed no effect on the intracellular α-syn accumulation in primary dopaminergic neurons. We were extremely curious about this compound’s effect on the α-syn accumulation in primary dopaminergic neurons since in the literature, both insulin and IGF-1R are reported to reduce α-syn toxicity in the neuroblastoma cell lines. Kao [53], for instance, has shown IGF-1 to block the formation of α-syn aggregates and suppress α-syn cytotoxicity in dopamine-treated SH-SY5Y cells; Chung and colleagues [54] hypothesized α-syn to play an important role in IGF-1 signaling activation in the neuroblastoma cell line. They suggested that α-syn co-regulates AKT activation: a lack of functional α-syn leads to decreased AKT phosphorylation, which in turn decreases the neuroprotective effect of IGF-1 in the brain. Such a discrepancy between our results and the above-mentioned studies may be due to the inherent limitations of the neuroblastoma cell line used for modeling dopamine neurons, including possible genetic abnormalities and only superficial resemblance to dopamine neurons [37]. Despite its drawbacks, neuroblastoma cell lines have been a very popular model, mainly due to their cost-effectiveness and ease of maintenance [55]. Primary neuronal cell cultures are physiologically more relevant, especially those rich in dopaminergic neurons, as in this study.

### 4.2. Formation of Intracellular α-Synuclein Accumulation in Dopaminergic Neurons Treated with Selective PTEN and SHIP2 Inhibitors

One, of course, should not discard the possibility that the concentration of the insulin present in DPM culturing media is insufficient to influence cell survival; thus, we observed no effect on the accumulation of misfolded α-syn upon inhibiting IRs and IGF-1Rs. Therefore, our next step was to investigate whether enhancing the activation of insulin signaling through the major insulin signaling pathway, PI3K-AKT-mTOR, might affect the survival of dopaminergic neurons and the formation of intracellular α-syn aggregates.

To enhance the PI3K-AKT-mTOR pathway, we treated midbrain cells with VO-OHpic, a very potent small-molecule PTEN inhibitor, active in nM concentrations. PTEN was initially identified as a tumor suppressor, an antagonist of the cell-growth-regulating AKT-mTOR signaling pathway [56]. PTEN has also been described to regulate insulin’s effect on glucose metabolism, and PTEN regulation has been linked to neurodegenerative disorders [57].

Ramalingam and Kim [29] have shown the importance of the PI3K/AKT pathway of insulin signaling, not only for neuronal survival in neurodegenerative diseases, but also for α-syn accumulation (for a review, see [58]). Kao [53] has shown that the α-syn cytotoxicity-reducing effect of IGF-1 is suppressed by the PI3K inhibitor, suggesting that the PI3K/AKT pathway plays a crucial role in α-syn accumulation and aggregation. Augmenting of the total and phosphorylated α-syn in the cells has been suggested to ensue from an increased level of GSK3β (one of the GSK3 isomers, negatively regulated by the PI3K/AKT signaling pathway) [59]. GSK3β plays an important role, not only in the expression and aggregation of α-syn, but also in promoting neurodegeneration by affecting neuroinflammation, mitochondrial dysfunction and oxidative stress [60]. Upon inhibition of PTEN and/or SHIP2, the PI3K/AKT pathway that negatively regulates GSK3β becomes activated, which leads to the launching of pro-survival mechanisms in the cells and the downregulation of GSK3β, followed by the decrease in the α-syn level.

The activation of the PI3K-AKT-mTOR signaling pathway was achieved by inhibiting the pathway negative modulators PTEN (VO-OHpic) and SHIP2 (AS1949490). Based on the previous research, we expected dopaminergic neurons treated with the PTEN inhibitor (VO-OHpic) to show better surviving patterns and a decrease in the accumulation of α-syn; however, the VO-OHpic treatment neither significantly affected TH-positive cell numbers nor pαSyn accumulation.

Similarly, the activation of PI3K-AKT-mTOR with SHIP2 inhibitor AS1949490 showed neither an increase in neuronal survival nor a significant reduction in pαSyn aggregates in dopaminergic neurons. Although Soeda and colleagues [48] described the effect of AS1949490 as an improvement of “the impairment of synaptic plasticity and memory formation in diabetic db/db mice” in the hippocampal slices in vitro in a concentration of 10 μM, treatment of the midbrain cell culture with 5 μM in the current study resulted in cell death.

One can speculate that, while there is a lack of a direct effect of insulin on dopaminergic neurons in vitro, there could still be some indirect effects in vivo through, for example, the action of insulin on glial cells.

Glial cells, and specifically astrocytes, could potentially play an indirect role in modulating the accumulation of aggregated α-syn in neurons, for example, by limiting pαSyn transmission [61,62]. Astrocytes were shown to be able to efficiently uptake and degrade pathological α-syn, thus protecting neurons in culture. Furthermore, glial cells can both produce GDNF, at least in the disease state, and express receptors for GDNF [63].

Astrocytes seem to be able to both secrete insulin and express insulin receptors [64]. It is possible that, in the brain, insulin could exert some protective effect against pαSyn accumulation through action on astrocytes—either by enhancing their ability to uptake and degrade pathological α-syn or by modulating GDNF release. However, in our study, we concentrated on the direct effects of insulin on dopamine neurons. Indirect, astrocyte-dependent effects of insulin would require further studies in dedicated astrocytes/neurons cocultures.

### 4.3. Formation of Intracellular α-Synuclein Aggregates in Dopaminergic Neurons Cultured in Different Media

Since neither the inhibition of IR/IGF-1R nor the facilitation of signaling through the PI3K/AKT pathway downstream to these receptors affected PFF-induced α-syn aggregation, we decided to investigate whether the complete withdrawal of insulin could affect this process.

Surprisingly, in dopaminergic neurons, a change of the culturing media to NB medium reduced the formation of PFF-induced α-syn aggregates more than twofold, while insulin removal in the cells cultured in NB-ins had no further effects. Factors influencing PFF-induced α-syn aggregates formation in primary neurons have already been discussed in the literature from the perspective of assay development [40]. The increased susceptibility of different neuronal populations—i.e., the higher rate of aggregate formation in hippocampal versus cortical neurons—have been reported [41], and this phenomenon was also clear in our experiment, with almost three times more aggregates in hippocampal than cortical cells. Literature data also show that more mature cultures are more susceptible to PFF-induced α-syn aggregate formation—possibly because of higher α-syn levels, linked with more developed neurites [40]. Our results demonstrated the previously unreported effect of the cell culture media formulation on α-syn aggregation. Crucially, the effect was evident in dopaminergic but not hippocampal and cortical neurons. Dopaminergic neurons, despite being of prime interest for modeling aspects of PD, have not actually been utilized in most studies and are often substituted with hippocampal and cortical cells, which are much easier to culture and obtain in large quantities. Our data illustrate the uniqueness of dopaminergic neurons in their response to culture conditions, suggesting that the formation of α-syn aggregates in primary dopaminergic cells may be more complex than in other cellular models. Thus, our results emphasize the importance of modeling α-syn aggregation in PD with cultures of primary dopaminergic neurons.

We have not investigated how the NB medium reduces the formation of α-syn aggregates in dopaminergic neurons. However, several possible mechanisms could be considered. The N2 supplement promotes the differentiation of postmitotic neurons in culture, while B27 supports survival and neurite outgrowth. All neurons were maintained for the first 5 days—a period when most spontaneous cell death in culture occurs—in the same media. However, it is still possible that switching to different culturing conditions at DIV5 promoted better survival of specific neuronal subpopulations. Another feasible explanation is that the media formulation can affect neurite outgrowth and, thus, α-syn levels. NB medium with a B27 supplement is more supportive for neurite growth. Thus, one would expect a rather more pronounced α-syn aggregation in this media. Furthermore, DPM vs. NB media could have different effects on the differentiation and survival of astrocytes, which may also affect PFF-induced α-syn aggregation [61,62]. Another possibility is that components in the NB medium activate specific pathways, similar to what was shown by us in regard to the inclusion of GDNF in culture media [30]. The observed effects of NB medium on survival might also have resulted in the preferential survival of a subpopulation of dopamine neurons, leading to a somewhat different neuronal composition of the cultures.

Regardless of the differences in α-syn aggregate formation in NB medium, withdrawal of insulin in NB-ins medium demonstrated no effect in this study, thus corroborating our results from the inhibition of IGF-1R/IR, SHIP2 and PTEN.

## 5. Conclusions

The main conclusions of our study are that insulin and insulin-related signaling does not influence PFF-induced α-syn aggregation in the primary dopamine neurons. Both SHIP2 inhibitor AS1949490 and PTEN inhibitor VO-OHpic failed to show a significant effect on α-syn aggregation, similar to IGF-1R/IR inhibitor GSK1904529A and to insulin withdrawal. In other projects [30,38,45], we have tested molecules unrelated to the insulin signaling pathway, which demonstrated an attenuation of α-syn aggregation in this model, hence, validating the approach. A second important finding of this study is that the choice of a culturing media can be crucial for the in vitro experiment—specifically the choice of the culturing media for primary dopaminergic neurons, but not hippocampal and cortical cells—as it can significantly affect the susceptibility of these cells to α-syn aggregation. This emphasizes both the importance of modeling α-syn aggregation in dopaminergic neurons and the need for considering the culturing conditions during assay development.

## Figures and Tables

**Figure 1 biomolecules-12-00563-f001:**
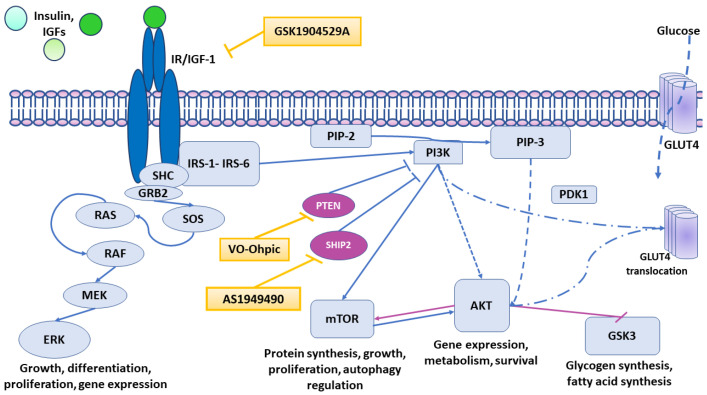
Schematic illustration of the actions of AS1949490, VO-Ohpic and GSK1904529A on downstream signaling of the insulin receptor. GSK1904529A selectively blocks IR and IGF1R autophosphorylation, in this way blocking the downstream signaling of IR and IGF1R. VO-Ohpic inhibits PTEN, thereby upregulating PI3K-AKT-mTOR signaling. AS1949490 increases the signaling through the PI3K-AKT-mTOR pathway by selectively inhibiting SHIP2.

**Figure 2 biomolecules-12-00563-f002:**
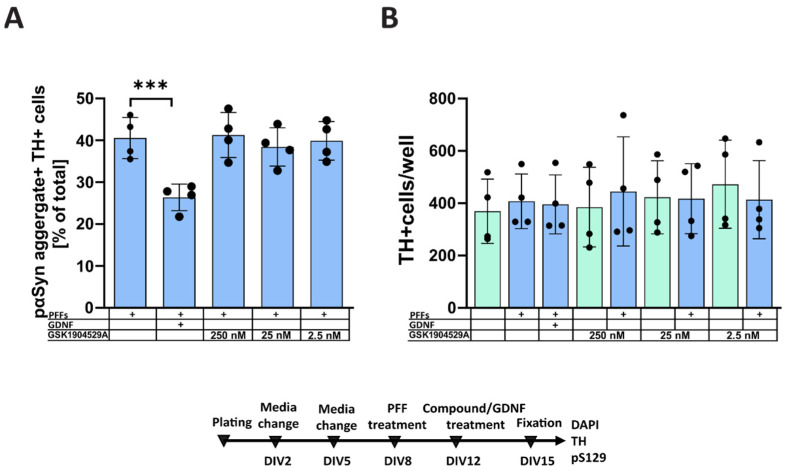
The effect of the selective IR and IGF-1R inhibitor GSK1904529A on α-syn aggregation in primary midbrain dopaminergic neurons, treated with PFFs. (**A**) Quantification of pαSyn-positive accumulations within TH-positive dopaminergic neurons. GDNF added to the cells on DIV12, the same day as a compound, significantly reduced the aggregations of intracellular pαSyn, while GSK1904529 showed no effect. (**B**) Quantification of TH-positive cells per well showed no significant changes, suggesting non-toxicity of the chosen concentrations of GSK1904529A to TH-positive dopaminergic cells. n = 4 independent experiments, data are mean ± SD, *** *p* < 0.001.

**Figure 3 biomolecules-12-00563-f003:**
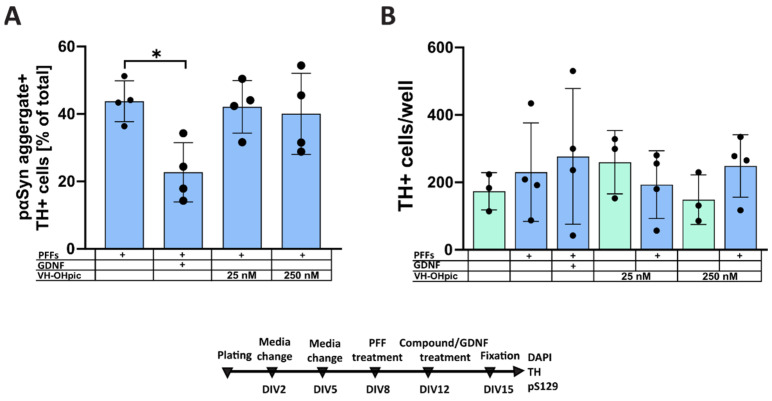
Effect of PTEN inhibitor VO-OHpic on the accumulation of pαSyn in midbrain dopaminergic neurons. Enhancing insulin signaling with PTEN inhibitor VO-OHpic had no significant effect on the accumulation of pαSyn in midbrain dopaminergic neurons. (**A**) Quantification of pαSyn-positive accumulations within TH-positive dopaminergic neurons. GDNF added to the cells on DIV12, the same day as a compound, significantly reduced the aggregations of intracellular pαSyn, while VO-OHpic showed no significant effect. (**B**) Quantification of TH-positive cells per well. n = 3–4 independent experiments, data are mean ± SD, * *p* < 0.05.

**Figure 4 biomolecules-12-00563-f004:**
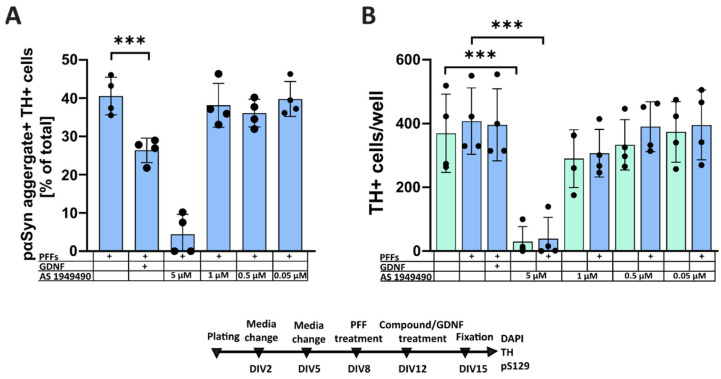
Effect of SHIP2 inhibitor AS1949490 on the accumulation of pαSyn in midbrain dopaminergic neurons. Enhancing insulin signaling with SHIP2 inhibitor AS1949490 exerted no significant impact on the accumulation of pαSyn in midbrain dopaminergic neurons in concentrations that have not affected neuron survival. (**A**) Quantification of pαSyn-positive accumulations within TH-positive dopaminergic neurons. GDNF added to the cells on DIV12, the same day as a compound, significantly reduced the aggregation of intracellular pαSyn, while AS1949490 showed no similar effect at any studied concentration. Note: AS1949490 in a concentration of 5 μM shows a significant decrease in TH-positive cell vitality, making interpretation of its effects on α-syn accumulation at this concentration unfeasible. (**B**) The number of TH-positive cells per well AS1949490 in a concentration of 5 µM proved to be toxic to the dopaminergic neurons of primary embryonic midbrain cell culture (two-way repeated-measures ANOVA, followed by Tukey’s multiple comparison test, *** *p* < 0.001, n = 4 independent experiments, data are mean ± SD).

**Figure 5 biomolecules-12-00563-f005:**
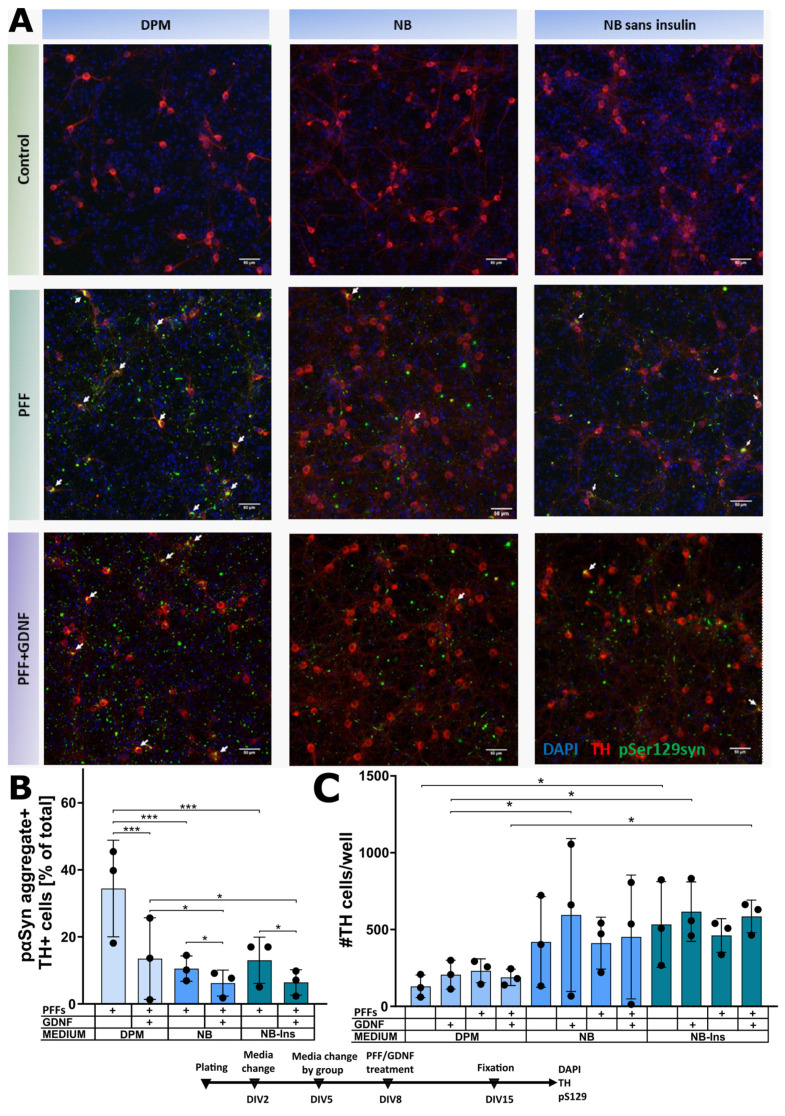
The effect of different culturing media (DPM, NB and NB-ins) on pαSyn aggregation in primary mouse embryonic dopaminergic neurons. (**A**) Representative images of control cells in DPM, NB and NB sans insulin (top), PFF-treated cells in DPM, NB and NB-ins (middle) and PFF- and GDNF-treated cells in DPM, NB and NB-ins. Scale bars 50 µm. (**B**) The effect of the media on the formation of intracellular LB-like inclusion in dopaminergic neurons of midbrain cell culture. (**C**) The number of TH-positive cells per well in correspondent media. n = 3 independent experiments, data are mean ± SD, *** *p* < 0.001, * *p* < 0.05.

**Figure 6 biomolecules-12-00563-f006:**
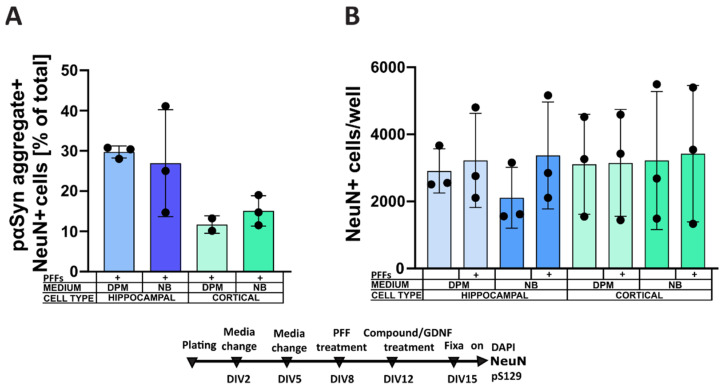
The effect of different culturing media (DPM, NB) on α-syn aggregation in primary mouse embryonic hippocampal (HIP) and cortical (CORT) neurons. (**A**) The effect of the media on the formation of intracellular LB-like inclusion in hippocampal and cortical neurons. (**B**) Number of NeuN-positive cells per well in correspondent media in hippocampal and cortical cultures. n = 3 independent experiments, data are mean ± SD.

## Data Availability

The data presented in this study are available upon reasonable request from the corresponding author.

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
