# Peer review of "Cell Culture Media, Unlike the Presence of Insulin, Affect α-Synuclein Aggregation in Dopaminergic Neurons"

_biomolecules, 2022, doi:10.3390/biom12040563_

Round 1
Reviewer 1 Report
This paper contravened the direct involvement of insulin signaling in the modulation of α-synuclein aggregation in dopamine neurons. They performed that the choice of culturing media can significantly affect preformed fibril-induced α-synuclein phosphorylation in primary dopaminergic cell culture.
I recommend this manuscript needs some revision before publication.
- Where is the GDNF from in PD? Why did that GDNF reduce the aggregation of intracellular pαSyn significantly?
- When the α-syn phosphorylation in the neuron, which is the main downstream signaling inside the neuron?
- The mechanism of the effects that different culture media was associated with the accumulation of α-syn didn’t explain well, the GDNF significant decreased the accumulation of α-syn, what’s the relationship between the GDNF and insulin? And what’s the GDNF downstream signaling?
- The effect of GDNF was specific to the dopamine neuron or is the same to glia cell? If add the glia culture medium without the insulin, did it produce the same effect in neuron?
Author Response
We thank the Reviewer for the comments on our manuscript and interesting inquiries about the GDNF. We have responded to these inquires below. However, in the manuscript we have kept the discussion on GDNF concise to avoid shifting the focus. We also thank you for your insightful comments regarding the possible involvement of glial cells. Such considerations have been now incorporated into the manuscript.
Please see the attached file ("Response to Reviewer 1.docx") for detailed response and revised manuscript.

Reviewer 2 Report
Hlushchuk et al investigate the role of the insulin signaling pathways on the intracellular accumulation of phosphorylated α-syn.
The paper is interesting and well structured. I was particularly positive about the discussion, which is level-headed and actually highlights the significance of the results obtained by the authors in comparison to other studies. The results presented may not represent a groundbreaking discovery, but the research is well justified
The manuscript needs minor editorial corrections, e.g. line 45 lacks spaces before citation.
What I have the biggest complaint about is the figures. The axis captions are partially cut, but the worst are the bold lines used in the tables under the graphs. They focus all the attention of the reader, also at first glance you cannot see anything but thick lines. The figures would be much more readable if the authors improved this element. Maybe a bold font would not have such a negative effect
Author Response
Response to reviewer 2:
We thank the reviewer 2 for the kind review and noticing problems with readability of figures. We apologize for this issue – it must have happened while converting Word document to PDF file due to the use of vector graphics. We will make sure line thickness is correct in PDF version and figures are readable.
In addition to the above comments, grammatical errors pointed out by the reviewers have been corrected.
We look forward to hearing from you in due time regarding our submission and to respond to any further questions and comments you may have.
Please see the revised version of the manuscript to see the corrected figures.
Round 2
Reviewer 1 Report
the authors had addressed all my concerns and I have no further